# Adversarial Boot Camp: label free certified robustness in one epoch

## Abstract

Machine learning models are vulnerable to adversarial attacks. One approach to addressing this vulnerability is certification, which focuses on models that are guaranteed to be robust for a given perturbation size. A drawback of recent certified models is that they are stochastic: they require multiple computationally expensive model evaluations with random noise added to a given image. In our work, we present a *deterministic* certification approach which results in a certifiably robust model. This approach is based on an equivalence between training with a particular regularized loss, and the expected values of Gaussian averages. We achieve certified models on ImageNet-1k by retraining a model with this loss for one epoch without the use of label information.

## 1 Introduction

Neural networks are very accurate on image classification tasks, but they are vulnerable to adversarial perturbations, i.e. small changes to the model input leading to misclassification (Szegedy et al., 2014). Adversarial training (Madry et al., 2018) improves robustness, at the expense of a loss of accuracy on unperturbed images (Zhang et al., 2019). Model certification (Lécuyer et al., 2019; Raghunathan et al., 2018; Cohen et al., 2019) is complementary approach to adversarial training, which provides a guarantee that a model prediction is invariant to perturbations up to a given norm.

Given an input $x$, the model $f$ is certified to $\ell_2$ norm $r$ at $x$ if it gives the same classification on $f(x + \eta)$ for all perturbation $\eta$ with norm up to $r$,

$$\arg\max f(x + \eta) = \arg\max f(x), \quad \text{for all } \|\eta\|_2 \leq r \tag{1}$$

Cohen et al. (2019) and Salman et al. (2019) certify models by defining a "smoothed" model, $f^{smooth}$, which is the expected Gaussian average of our initial model $f$ at a given input example $x$,

$$f^{smooth}(x) \approx \mathbb{E}_\eta \left[ f(x + \eta) \right] \tag{2}$$

where the perturbation is sampled from a Gaussian, $\eta \sim \mathcal{N}(0, \sigma^2 I)$. Cohen et al. (2019) used a probabilistic argument to show that models defined by (2) can be certified to a given radius by making a large number of stochastic model evaluations. Certified models can classify by first averaging the model, (Salman et al., 2019), or by taking the mode, the most popular classification given by the ensemble (Cohen et al., 2019).

Cohen et al. and Salman et al. approximate the model $f^{smooth}$ stochastically, using a Gaussian ensemble, which consists of evaluating the base model $f$ multiple times on the image perturbed by noise. Like all ensemble models, these stochastic models require multiple inferences, which is more costly than performing inference a single time. In addition, these stochastic models require training the base model $f$ from scratch, by exposing it to Gaussian noise, in order to improve the accuracy of $f^{smooth}$. Salman et al. (2019) additionally expose the model to adversarial attacks during training. In the case of certified models, there is a trade-off between certification and accuracy: the certified models lose accuracy on unperturbed images.

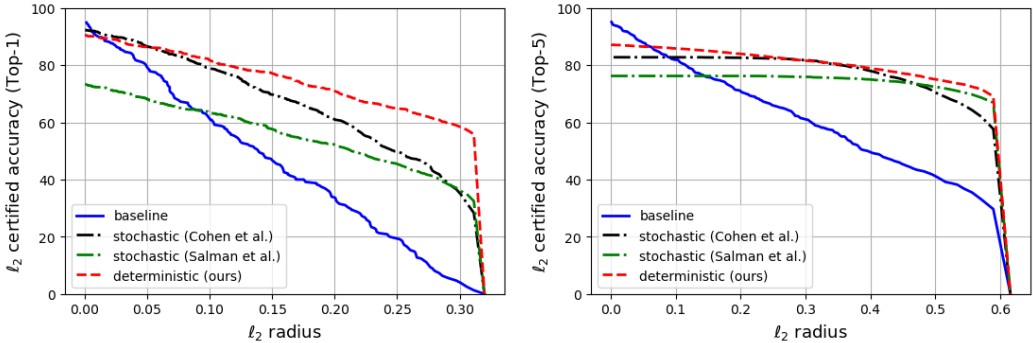

**(a)** *CIFAR-10 top-1 certified accuracy, $\sigma = 0.10$* **(b)** *ImageNet top-5 certified accuracy, $\sigma = 0.25$*

**Figure 1:** *Certified accuracy as a function of $\ell_2$ radius.*

In this work, we present a *deterministic* model $f^{smooth}(x)$ given by (2). Unlike the stochastic models, we do not need to train a new model from scratch with Gaussian noise data augmentations, instead, we can fine tune an accurate baseline model for one epoch, using a loss designed to encourage (2) to hold. The result is a certified deterministic model which is *faster to train, and faster to perform inference*. In addition, the certification radius for our model is improved, compared to previous work, on both the CIFAR-10 and ImageNet databases, see Figure 1. Moreover, the accuracy on unperturbed images is improved on ImageNet with a Top-5 accuracy of 86.1% for our deterministic model versus 85.3% for Cohen et al. (2019) and 78.8% for Salman et al. (2019).

To our knowledge, this is the first deterministic Gaussian smoothing certification technique. The main appeal of our approach is a large decrease in the number of function evaluations for computing certifiably robust models, and a corresponding decrease in compute time at inference. Rather than stochastically sampling many times from a Gaussian at inference time, our method is certifiably robust with only a single model query. This greater speed in model evaluation is demonstrated in Table 2. Moreover, the deterministic certified model can be obtained by retraining a model for one epoch, without using labels. The speed and flexibility of this method allows it to be used to make empirically robust models certifiably robust. To see this, we test our method on adversarially trained models from Madry et al. (2018), increasing the certified radius of the adversarially robust model, see Table 4.

**Table 1:** *A comparison of robust models. Stochastic smoothing arises from methods like the ones presented in Cohen et al. (2019) and Salman et al. (2019). Adversarial training from Madry et al. (2018).*

| Model | Can be obtained from any pretrained model | Evaluation in one forward pass | Is certified? |
|---|:---:|:---:|:---:|
| Deterministic Smoothing (ours) | ✔ | ✔ | ✔ |
| Stochastic Smoothing | ✗ | ✗ | ✔ |
| Adversarial Training | ✗ | ✔ | ✗ |

**Table 2:** *Average classification inference time (seconds)*

| Model | CIFAR-10 | | ImageNet-1k | |
|---|:---:|:---:|:---:|:---:|
| | CPU | GPU | CPU | GPU |
| Deterministic (ours) | 0.0049 | 0.0080 | 0.0615 | 0.0113 |
| Stochastic (Cohen et al., 2019) | 0.0480 | 0.0399 | 0.1631 | 0.0932 |

## 2 Related work

The issue of adversarial vulnerability arose in the works of Szegedy et al. (2014) and Goodfellow et al. (2015), and has spawned a vast body of research. The idea of training models to be robust to adversarial attacks was widely popularized in Madry et al. (2018). This method, known as *adversarial training*, trains a model on images corrupted by gradient-based adversarial attacks, resulting in robust models. In terms of certification, early work by Cheng et al. (2017) provided a method of computing maximum perturbation bounds for neural networks, and reduced to solving a mixed integer optimization problem. Weng et al. (2018a) introduced non-trivial robustness bounds for fully connected networks, and provided tight robustness bounds at low computational cost. Weng et al. (2018b) proposed a metric that has theoretical grounding based on Lipschitz continuity of the classifier model and is scaleable to state-of-the-art ImageNet neural network classifiers. Zhang et al. (2018) proposed a general framework to certify neural networks based on linear and quadratic bounding techniques on the activation functions, which is more flexible than its predecessors.

Training a neural network with Gaussian noise has been shown to be equivalent to gradient regularization (Bishop, 1995). This helps improve robustness of models; however, recent work has used additive noise during training and evaluation for certification purposes. Lécuyer et al. (2019) first considered adding random Gaussian noise as a certifiable defense in a method called *PixelDP*. In their method, they take a known neural network architecture and add a layer of random noise to make the model's output random. The expected classification is in turn more robust to adversarial perturbations. Furthermore, their defense is a certified defense, meaning they provide a lower bound on the amount of adversarial perturbations for which their defence will always work. In a following work, Li et al. (2018) provided a defence with improved certified robustness. The certification guarantees given in these two papers are loose, meaning the defended model will always be more robust than the certification bound indicates.

In contrast, Cohen et al. (2019) provided a defence utilizing randomized Gaussian smoothing that leads to *tight* robustness guarantees under the $\ell_2$ norm. Moreover Cohen et al. used Monte Carlo sampling to compute the radius in which a model's prediction is unchanged; we refer to this method as RANDOMIZEDSMOOTHING. In work building on Cohen et al., Salman et al. (2019) developed an adversarial training framework called SMOOTHADV and defined a Lipschitz constant of averaged models. Yang et al. (2020) generalize previous randomized smoothing methods by providing robustness guarantees in the $\ell_1$, $\ell_2$, and $\ell_\infty$ norms for smoothing with several non-Gaussian distributions.

## 3 Deterministic Smoothing

Suppose we are given a dataset consisting of paired samples $(x, y) \in \mathcal{X} \times \mathcal{Y}$ where $x$ is an example with corresponding true classification $y$. The supervised learning approach trains a model $f : \mathcal{X} \longrightarrow \mathbb{R}^{Nc}$ which maps images to a vector whose length equals the number of classes. Suppose $f$ is the initial model, and let $f^{smooth}$ be the averaged model given by equation (2). Cohen et al. (2019) find a Gaussian smoothed classification model $f^{smooth}$ by sampling $\eta \sim \mathcal{N}(0, \sigma^2 I)$ independently $n$ times, performing $n$ classifications, and then computing the most popular classification. In the randomized smoothing method, the initial model $f$ is trained on data which is augmented with Gaussian noise to improve accuracy on noisy images.

We take a different approach to Gaussian smoothing. Starting from an accurate pretrained model $f$, we now discard the training labels, and iteratively retrain a new model, $f^{smooth}$ using a quadratic loss between the model $f$ and the new model's predictions, with an additional gradient regularization term. We have found that discarding the original one-hot labels and instead using model predictions helps make the model smoother.

To be precise, our new models is a result of minimizing the loss which we call HEATSMOOTH-ING,

$$\mathbb{E}_x \left[ \frac{1}{2} \left\| \text{softmax} \left( f^{smooth}(x) \right) - \text{softmax} \left( f(x) \right) \right\|_2^2 + \frac{\sigma^2}{2} \left\| \nabla_x f^{smooth}(x) \right\|_2^2 \right] \qquad (3)$$

The smoothing achieved by the new models is illustrated schematically in Figure 5.

### 3.1 Related regularized losses

Gradient regularization is known to be equivalent to Gaussian smoothing (Bishop, 1995; LeCun et al., 1998). Our deterministic smoothed model arises by training using the HEATSMOOTHING loss (3), which is designed so to ensure that (2) holds for our model. Our results is related to the early results on regularized networks (Bishop, 1995; LeCun et al., 1998): that full gradient regularization is equivalent to Gaussian smoothing. Formally this is stated as follows.

**Theorem 1.** *(Bishop, 1995) Training a feed-forward neural-network model using the quadratic (or mean-squared error) loss, with added Gaussian noise of mean 0 and variance $\sigma^2$ to the inputs, is equivalent to training with*

$$\mathbb{E}_x \left[ \|f(x) - y\|^2 + \sigma^2 \|\nabla f(x)\|^2 \right] \tag{4}$$

*up to higher order terms.*

The equivalence is normally used to go from models augmented with Gaussian noise to regularized models. In our case, we use the result in the other direction: we train a regularized model in order to produce a model which is equivalent to evaluating with noise. In practice, this means that rather than adding noise to regularize models for certifiable robustness, we explicitly perform a type of gradient regularization, *in order to produce a model which performs as if Gaussian noise was added.* See Figure 4 in Appendix D for an illustration of the effect of this gradient regularization.

The gradient regularization term in the HEATSMOOTHING loss (3), is also related to adversarial training. Tikhonov regularization is used to produced adversarially robust models (Finlay and Oberman, 2019). However in adversarial training, the gradient of the loss is used, rather that the gradient of the full model. Also, our loss does not use information from the true labels. The reason for these differences is due to the fact that we wish to have a model that approximates the Gaussian average of our initial model $f$ (see Appendix A). Furthermore, minimizing the gradient-norm of the loss of the output gives us a smooth model in all directions, rather than being robust to only adversarial perturbations.

### 3.2 Algorithmic Details

We have found that early on in training, the value $\frac{1}{2} \left\| f^{smooth}(x) - f^k(x) \right\|_2^2$ may be far greater than the $\frac{\sigma^2}{2} \left\| \nabla_x f^{smooth}(x) \right\|_2^2$ term. So we introduced a softmax of the vectors in the distance-squared term to reduce the overall magnitude of this term. We perform the training minimization of (3) for one epoch. The pseudo-code for our neural network weight update is given by Algorithm 1 [1]

Note that the $\left\| \nabla_x f^{smooth}(x) \right\|_2^2$ term in (3) requires the computation of a Jacobian matrix norm. In high dimensions this is computationally expensive. To approximate this term, we make use of the *Johnson-Lindenstrauss lemma* (Johnson and Lindenstrauss, 1984; Vempala, 2005) followed by the finite difference approximation from Finlay and Oberman (2019). We are able to approximate $\left\| \nabla_x f^{smooth}(x) \right\|_2^2$ by taking the average of the product of the Jacobian matrix and Gaussian noise vectors. Jacobian-vector products can be easily computed via reverse mode automatic differentiation, by moving the noise vector $w$ inside:

$$w \cdot (\nabla_x v(x)) = \nabla_x (w \cdot v(x)) \tag{5}$$

Further computation expense is reduced by using finite-differences to approximate the norm of the gradient. Once the finite-difference is computed, we detach this term from the automatic differentiation computation graph, further speeding training. More details of our implementation of these approximation techniques, and the definition of the term $\hat{g}$ which is a regularization of the gradient, are presented in Appendix B.

---

**Algorithm 1:** HEATSMOOTHING Neural Network Weight Update

---

**Input** : Minibatch of input examples $\boldsymbol{x}^{(mb)} = \left(x^{(1)}, \ldots, x^{(Nb)}\right)$
A model $v$ set to "train" mode
Current model $f$ set to "eval" mode
$\sigma$, standard deviation of Gaussian smoothing
$\kappa$, number of Gaussian noise replications (default= 10)
$\delta$, finite difference step-size (default= 0.1)

**Update** : learning-rate according to a pre-defined scheduler.

**for** $i \in \{1, \ldots Nb\}$ **do**

$\quad$ **Compute**: $f^{smooth}(x^{(i)}), f(x^{(i)}) \in \mathbb{R}^{Nc}$

$\qquad\qquad J_i = \frac{1}{2} \left\| f^{smooth}(x^{(i)}) - f(x^{(i)}) \right\|_2^2 \in \mathbb{R}$

$\qquad\qquad$ **for** $j \in \{1, \ldots \kappa\}$ **do**

$\qquad\qquad\qquad$ Generate $w = \frac{1}{\sqrt{Nc}} \left(w_1, \ldots, w_{Nc}\right), w_1, \ldots, w_{Nc} \in \mathcal{N}(0, 1)$

$\qquad\qquad\qquad$ Compute the normalized gradient $\hat{g}$ via (18)

$\qquad\qquad\qquad$ Detach $x^{(i)}$ from the computation graph

$\qquad\qquad\qquad J_i \leftarrow J_i + \frac{\sigma^2}{2\delta^2} \left(w \cdot f^{smooth}(x^{(i)} + \delta \hat{g}) - w \cdot f^{smooth}(x^{(i)})\right)^2$

$\qquad\qquad$ **end**

$\quad J \leftarrow \frac{1}{Nb} \sum\limits_{i=1}^{Nb} J_i$

**end**

Update the weights of $v$ by running backpropagation on $J$ at the current learning rate.

---

## 3.3 THEORETICAL DETAILS

We appeal to partial differential equations (PDE) theory for explaining the equivalence between gradient regularization and Gaussian convolution (averaging) of the model[2]. The idea is that the gradient term which appears in the loss leads to a smoothing of the new function (model). The fact that the exact form of the smoothing corresponds to Gaussian convolution is a mathematical result which can be interpreted probabilistically or using techniques from analysis. Briefly, we detail the link as follows.

Einstein (1906) showed that the function value of an averaged model under Brownian motion is related to the heat equation (a PDE); the theory of stochastic differential equations makes this rigorous (Karatzas and Shreve, 1998). Moreover, solutions of the heat equation are given by Gaussian convolution with the original model. Crucially, in addition solutions of the heat equation can be interpreted as iterations of a regularized loss problem (called a variational energy) like that of equation 3. The minimizer of this variational energy (3) satisfies an equation which is formally equivalent to the heat equation (Gelfand et al., 2000). Thus, taking these facts together, we see that a few steps of the minimization of the loss in (3) yield a model which approximately satisfies the heat equation, and corresponds to a model smoothed by Gaussian convolution. See Figure 4 for an illustration of a few steps of the training procedure. This result is summarized in the following theorem.

**Theorem 2.** *(Strauss, 2007) Let $f$ be a bounded function, $x \in \mathbb{R}^d$, and $\eta \sim \mathcal{N}\left(0, \sigma^2 I\right)$. Then the following are equivalent:*

1. $\mathbb{E}_\eta \left[f(x + \eta)\right]$, *the expected value of Gaussian averages of $f$ at $x$.*

2. $\left(f * \mathcal{N}(0, \sigma^2 I)\right)(x)$, *the convolution of $f$ with the density of the $\mathcal{N}(0, \sigma^2 I)$ distribution evaluated at $x$.*

---

[1]Code and links to trained models are posted in Supplemental Materials

[2]We sometimes interchange the terms Gaussian averaging and Gaussian convolution; they are equivalent, as shown in Theorem 2.

3. *The solutions of the heat equation,*

$$\frac{\partial}{\partial t} f(x,t) = \frac{\sigma^2}{2} \Delta_x f(x,t) \tag{6}$$

*at time $t = 1$, with initial condition $f(x,0) = f(x)$.*

In Appendix A, we use Theorem 2 to show the equivalence of training with noise and iteratively training (3).

To assess how well our model approximates the Gaussian average of the initial model, we compute the certified $\ell_2$ radius for averaged models introduced in Cohen et al. (2019). A larger radius implies a better approximation of the Gaussian average of the initial model. We compare our models with stochastically averaged models via *certified accuracy*. This is the fraction of the test set which a model correctly classifies at a given radius while ignoring abstained classifications. Throughout, we always use the same $\sigma$ value for certification as for training. In conjunction with the certification technique of Cohen et al., we also provide the following theorem, which describes a bound based on the Lipschitz constant of a Gaussian averaged model. We refer to this bound as the $L$-bound, which demonstrates the link between Gaussian averaging and adversarial robustness.

**Theorem 3** ($L$-**bound**). *Suppose $f^{smooth}$ is the convolution (average) of $f : \mathbb{R}^d \to [0,1]^{Nc}$ with a Gaussian kernel of variance $\sigma^2 I$,*

$$f^{smooth}(x) = \left( f * \mathcal{N}(0, \sigma^2 I) \right)(x)$$

*Then any perturbation $\delta$ which results in a change of rank of the k-th component of $f^{smooth}(x)$ must have norm bounded as follows:*

$$\|\delta\|_2 \geq \sigma(\pi/2)^{1/2}(f^{smooth}(x)_{(k)} - f^{smooth}(x)_{(k+1)}) \tag{7}$$

*where $f^{smooth}(x)_{(i)}$ is the $i^{th}$ largest value in the vector $f^{smooth}(x) \in [0,1]^{Nc}$.*

See Appendix C for proof. This bound is equally applicable to deterministic or stochastically averaged models. In stochastically averaged models $f^{smooth}(x)$ is replaced by the stochastic approximation of $\mathbb{E}_{\eta \sim \mathcal{N}(0,\sigma^2 I)} [f(x + \eta)]$.

## 3.4 Adversarial Attacks

To test how robust our model is to adversarial examples, we calculate the minimum $\ell_2$ adversarial via our $L$-bound and we attack our models using the *projected gradient descent (PGD)* (Kurakin et al., 2017; Madry et al., 2018) and *decoupled direction and norm (DDN)* (Rony et al., 2019) methods. These attacks are chosen because there is a specific way they can be applied to stochastically averaged models (Salman et al., 2019). In the $\ell_2$ setting of both attacks, it is standard to take the step

$$g = \alpha \frac{\nabla_{\delta_t} L\left(f(x + \delta_t), y\right)}{\|\nabla_{\delta_t} L\left(f(x + \delta_t), y\right)\|_2} \tag{8}$$

in the iterative algorithm. Here, $x$ is an input example with corresponding true class $y$; $\delta_t$ denotes the adversarial perturbation at its current iteration; $L$ denotes the cross-entropy Loss function (or KL Divergence); $\varepsilon$ is the maximum perturbation allowed; and $\alpha$ is the step-size. In the stochastically averaged model setting, the step is given by

$$g_n = \alpha \frac{\sum\limits_{i=1}^{n} \nabla_{\delta_t} L\left(f(x + \delta_t + \eta_i), y\right)}{\left\|\sum\limits_{i=1}^{n} \nabla_{\delta_t} L\left(f(x + \delta_t + \eta_i), y\right)\right\|_2} \tag{9}$$

where $\eta_1, \ldots, \eta_n \overset{\text{iid}}{\sim} \mathcal{N}(0, \sigma^2 I)$. For our deterministically averaged models, we implement the update (8). This is because our models are deterministic, meaning there is no need to sample noise at evaluation time. For stochastically averaged models (Cohen et al., 2019; Salman et al., 2019), we implement the update (9).

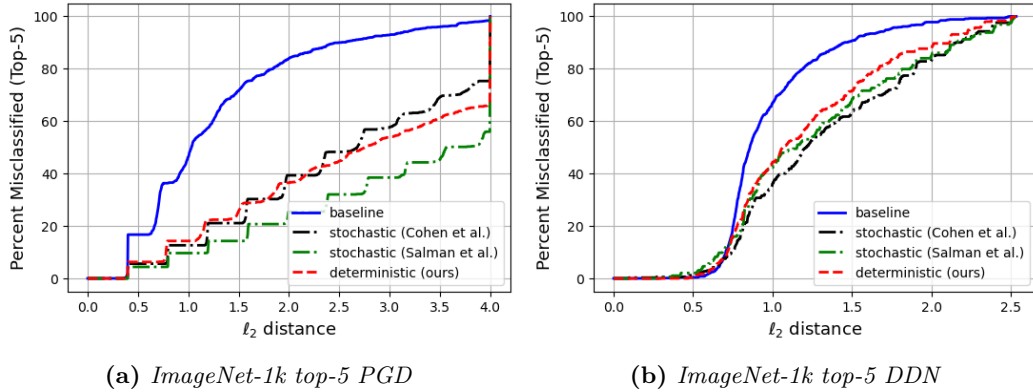

**(a)** *ImageNet-1k top-5 PGD*  **(b)** *ImageNet-1k top-5 DDN*

**Figure 2:** *Attack curves: % of images successfully attacked as a function of $\ell_2$ adversarial distance.*

## 4  EXPERIMENTS & RESULTS

We now execute our method on the ImageNet-1k dataset (Deng et al., 2009) with the ResNet-50 model architecture. The initial model $f$ was trained on clean images for 29 epochs with the cross-entropy loss function. Due to a lack of computing resources, we had to modify the training procedure (3) and Algorithm 1 to obtain our smoothed model $f^{smooth}$. This new training procedure amounts to minimizing the loss function

$$\frac{1}{2} \left\| \text{softmax} \left( f^{smooth}(x + \eta) \right) - \text{softmax} \left( f(x) \right) \right\|_2^2 + \frac{\sigma^2}{2} \left\| \nabla_x f^{smooth}(x + \eta) \right\|_2^2 \qquad (10)$$

for only 1 epoch of the training set using stochastic gradient descent at a fixed learning rate of 0.01 and with $\sigma = 0.25$. This is needed because the output vectors in the ImageNet setting are of length 1,000. Using softmax in the calculation of the $\ell_2$ distance metric prevents the metric from dominating the gradient-penalty term and the loss blowing up. Furthermore, we add noise $\eta \sim \mathcal{N}\left(0, \sigma^2 I\right)$ to half of the training images.

### 4.1  COMPARISON TO STOCHASTIC METHODS VIA CERTIFIED RADII

We compare our results to a pretrained RANDOMIZEDSMOOTHING ResNet-50 model with $\sigma = 0.25$ provided by Cohen et al. (2019). We also compare to a pretrained SMOOTHADV ResNet-50 model trained with 1 step of PGD and with a maximum perturbation of $\varepsilon = 0.5$ with $\sigma = 0.25$ provided by Salman et al. (2019). To assess certified accuracy, we run the CERTIFY algorithm from Cohen et al. (2019) with $n_0 = 25, n = 1,000, \sigma = 0.25$ for the stochastically trained models. We realize that this may not be an optimal number of noise samples, but it was the most our computational resources could handle. For the HEATSMOOTHING model, we run the same certification algorithm but without running SAMPLINGUNDERNOISE to compute $\hat{c}_A$. For completeness, we also certify the baseline model $f^0$. Certification results on 5,000 ImageNet test images are presented in Figure 1b. We see that our model is indeed comparable to the stochastic methods presented in earlier paper, despite the fact that we only needed one training epoch. Note that CIFAR-10 results are presented in Appendix E.

**Table 3:** *$\ell_2$ adversarial distance metrics on ImageNet-1k*

| Model | L-bound | | PGD | | DDN | |
|---|---|---|---|---|---|---|
| | median | mean | median | mean | median | mean |
| HEATSMOOTHING | 0.240 | 0.190 | 2.7591 | 2.6255 | 1.0664 | 1.2261 |
| SMOOTHADV | 0.160 | 0.160 | 3.5643 | 3.0244 | 1.1537 | 1.2850 |
| RANDOMIZEDSMOOTHING | 0.200 | 0.180 | 2.6787 | 2.5587 | 1.2114 | 1.3412 |
| Undefended baseline | - | - | 1.0313 | 1.2832 | 0.8573 | 0.9864 |

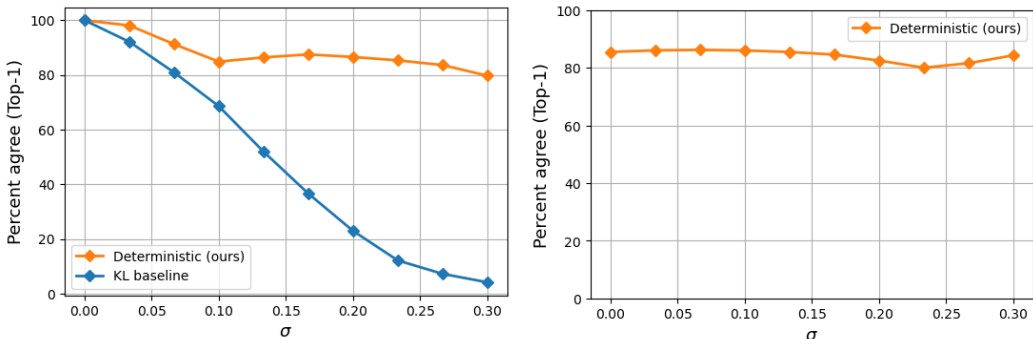

**(a)** *top-1 classification agreement between a given model and its own stochastic average prediction for varying σ.*

**(b)** *top-1 classification agreement between our deterministic model and a Cohen et al. model prediction for varying σ.*

**Figure 3**

### 4.2 Comparison to Stochastic Methods via Adversarial Attacks

Next, we attack our four models using PGD and DDN. For the stochastic models, we do 25 noise samples to compute the loss. We run both attacks with max $\ell_2$ perturbation of $\epsilon = 4.0$ until top-5 misclassification is achieved or until 20 steps are reached. Results on 1,000 ImageNet test images are presented in Table 3 and Figures 2a and 2b. We see that our model is comparable to the stochastic models, but does not outperform them. In Figure 2a, it is clear that the model presented in Salman et al. (2019) performs best, since this model was trained on PGD-corrupted images. Note that CIFAR-10 results are presented in Appendix E.

### 4.3 Comparing Models Using Their Classifications

Recall that the goal of this work is to use deterministic methods to obtain an averaged model equivalent to that of Cohen et al. (2019). One way of measuring the similarity is to compare the predictions of both models. To do this, we consider a pretrained stochastically averaged model with $\sigma = 0.25$ and our deterministic model trained using (10). We randomly select $1,000$ ImageNet test images and assess the difference between evaluating our model in a single forward pass vs. the stochastic method presented in Cohen et al.. In Figure 3a, we see how often a model's isnge forward pass top-1 prediction matches with predictions of this model using Cohen et al.'s PREDICT algorithm, conditioning on the single forward pass prediction being correct. In Figure 3b, we then see how often our deterministic model agrees with Cohen et al.'s stochastic model's prediction. In these plots, for stochastic prediction, we fix the number of Gaussian replications $n = 100$. We see that our model's predictions are the same when doing a single forward pass and doing a stochastic prediction with high probability. We also see that our model's deterministic predictions match a high-performing stochastically averaged model's predictions with high probability.

### 4.4 Certifying Robust Models

So far, we have showed that we can take a non-robust baseline model and make it certifiably robust by retraining for one epoch with a regularized loss (10). A natural question arises: can we use this method to make robust models certifiably robust? To test this, we begin with an adversarially trained model (Madry et al., 2018). This pretrained model was downloaded from Madry's "Robustness" GitHub repository and was trained with images corrupted by the $L_2$ PGD attack with maximum perturbation size $\epsilon = 3.0$. We certify this model by retraining it with (10) for one epoch using stochastic gradient descent with fixed learning rate 0.01. In Table 4, we compute the $\ell_2$ certified radius from Cohen et al. (2019) for these models using 1,000 ImageNet-1k test images with $\sigma = 0.25$. The certified radii for the model trained with the loss function (10) are significantly higher than those of the adversarially trained model from Madry et al. (2018).

**Table 4:** *$\ell_2$ certified radii summary statistics for robust models on ImageNet-1k*

| Model | $\ell_2$ radius | | |
|---|---|---|---|
| | median | mean | max. |
| Certified adversarially trained | 0.4226 | 0.4193 | 0.6158 |
| Adversarially trained | 0.0790 | 0.1126 | 0.6158 |
| Undefended baseline | 0.0 | 0.1446 | 0.6158 |

## 5 CONCLUSION

Randomized smoothing is a well-known method to achieve a Gaussian average of some initial neural network. This is desirable to guarantee that a model's predictions are unchanged given perturbed input data. In this work, we used a regularized loss to obtain deterministic Gaussian averaged models. By computing $\ell_2$ certified radii, we showed that our method is comparable to previously-known stochastic methods. This is confirmed by attacking our models, which results in adversarial distances similar to those seen with stochastically smoothed models. We also developed a new lower bound on perturbations necessary to throw off averaged models, and used it as a measure of model robustness. Lastly, our method is less computationally expensive in terms of inference time (see Table 2).

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

# A  Solving the heat equation by training with a regularized loss function

Theorem 2 tells us that training a model with added Gaussian noise is equivalent to training a model to solve the heat equation. We can discretize the heat equation (6) to obtain

$$\frac{f^{k+1} - f^k}{h} = \frac{\sigma^2}{2} \Delta f^{k+1} \tag{11}$$

for $k = 0, \ldots, n_T - 1$, where $n_T$ is the fixed number of timesteps, $h = 1/n_T$, and $f^0 = f$, our initial model. Notice how, using the Euler-Lagrange equation, we can express $f^{k+1}$ in (11) as the variational problem

$$f^{k+1} = \operatorname*{argmin}_{v} \frac{1}{2} \int_{\mathbb{R}^d} \left( \left| v(x) - f^k(x) \right|^2 + \frac{h\sigma^2}{2} \left\| \nabla_x v(x) \right\|_2^2 \right) \rho(x) dx \tag{12}$$

where $\rho$ is the density from which our clean data comes form. Therefore, this is equivalent to solving

$$f^{k+1} = \operatorname*{argmin}_{v} \mathbb{E}_x \left[ \left| v(x) - f^k(x) \right|^2 + \frac{h\sigma^2}{2} \left\| \nabla_x v(x) \right\|_2^2 \right] \tag{13}$$

Note that the minimizer of the objective of (12) satisfies

$$v - f^k = \frac{h\sigma^2}{2} \Delta v \tag{14}$$

which matches (11) if we set $f^{k+1} = v$. In the derivation of (14), we take for granted the fact that empirically, $\rho$ is approximately uniform and is therefore constant. In the end, we iteratively compute (13) and obtain models $f^1, \ldots, f^{n_T}$, setting $v = f^{n_T}$, our smoothed model.

Something to take note of is that our model outputs be vectors whose length corresponds to the total number of classes; therefore, the objective function in (13) will not be suitable for vector-valued outputs $f^k(x)$ and $v(x)$. We instead use the following update

$$f^{k+1} = \operatorname*{argmin}_{v} \mathbb{E}_x \left[ \frac{1}{2} \left\| v(x) - f^k(x) \right\|_2^2 + \frac{h\sigma^2}{2} \left\| \nabla_x v(x) \right\|_2^2 \right] \tag{15}$$

# B  Approximating the gradient-norm regularization term

By the Johnson-Lindenstrauss Lemma (Johnson and Lindenstrauss, 1984; Vempala, 2005), $\left\| \nabla_x v(x) \right\|_2^2$ has the following approximation,

$$\begin{aligned}
\left\| \nabla_x v(x) \right\|_2^2 &\approx \sum_{i=1}^{\kappa} \left\| \nabla_x \left( w_i \cdot v(x) \right) \right\|_2^2 \\
&\approx \sum_{i=1}^{\kappa} \left( \frac{\left( w_i \cdot v\left( x + \delta \hat{g}_i \right) \right) - \left( w_i \cdot v(x) \right)}{\delta} \right)^2
\end{aligned} \tag{16}$$

where

$$w_i = \frac{1}{\sqrt{K}} \left( w_{i1}, \ldots, w_{iK} \right)^T \in \mathbb{R}^K , \quad w_{ij} \overset{\text{iid}}{\sim} \mathcal{N}(0, 1) \tag{17}$$

and $\hat{g}_i$ is given by

$$\hat{g}_i = \begin{cases} \frac{\nabla_x (w_i \cdot v(x))}{\left\| \nabla_x (w_i \cdot v(x)) \right\|_2} & \text{if } \nabla_x \left( w_i \cdot v(x) \right) \neq 0 \\ 0 & \text{otherwise} \end{cases} \tag{18}$$

In practice, we set $\delta = 0.1$, $\kappa = 10$, and $K = Nc$, the total number of classes.

## C  Proof of Theorem 3

*Proof.* Suppose the loss function $\ell$ is Lipschitz continuous with respect to model input $x$, with Lipschitz constant $L$. Let $\ell_0$ be such that if $\ell(x) < \ell_0$, the model is always correct. Then by Proposition 2.2 in Finlay and Oberman (2019), a lower bound on the minimum magnitude of perturbation $\delta$ necessary to adversarially perturb an image $x$ is given by

$$\|\delta\|_2 \geq \frac{\max\{\ell_0 - \ell(x), 0\}}{L} \tag{19}$$

By Lemma 1 of Appendix A in Salman et al. (2019), our averaged model

$$f^{smooth}(x) = \left(f * \mathcal{N}(0, \sigma^2 I)\right)(x)$$

has Lipschitz constant $L = \frac{1}{\sigma}\sqrt{\frac{2}{\pi}}$. Replacing $L$ in (19) with $\frac{1}{\sigma}\sqrt{\frac{2}{\pi}}$ and setting $\ell_0 = f^{smooth}(x)_{(k)}, \ell(x) = f^{smooth}(x)_{(k+1)}$ gives us the proof. $\qquad\square$

## D  Illustration of regularized training

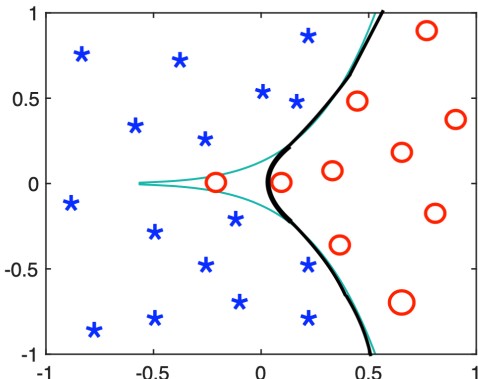

**Figure 4:** *Illustration of gradient regularization (4) in the binary classification setting. The lighter line represents classification boundary for original model with large gradients, and the darker line represents classification boundary of the smoothed model. The symbols indicate the classification by the original model: a single red circle is very close to many blue stars. The smoothed model has a smoother classification boundary which flips the classification of the outlier.*

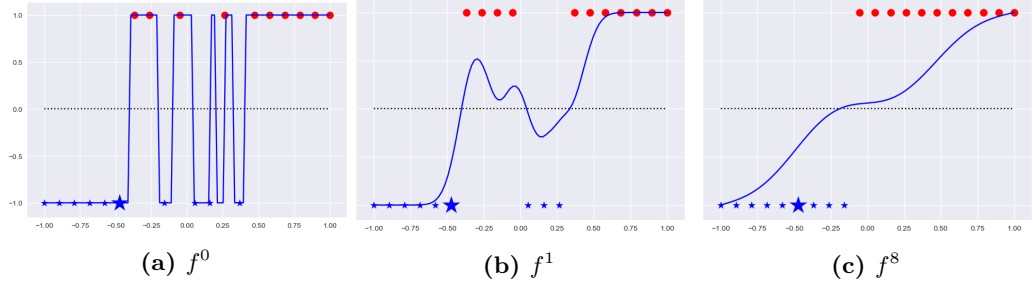

**(a)** $f^0$          **(b)** $f^1$          **(c)** $f^8$

**Figure 5:** *Illustration of performing the iterative model update (15) for 8 timesteps in the binary classification setting. The dashed black line represents our decision boundary. The blue line represents our current classification model. The blue stars and red circles represent our predicted classes using the current model iteration. Consider the datapoint at $x = -0.5$. In the initial model $f^0$, the adversarial distance is $\approx 0.10$. In model $f^5$, the adversarial distance is increased to $\approx 0.35$.*

# E    RESULTS ON THE CIFAR-10 DATASET

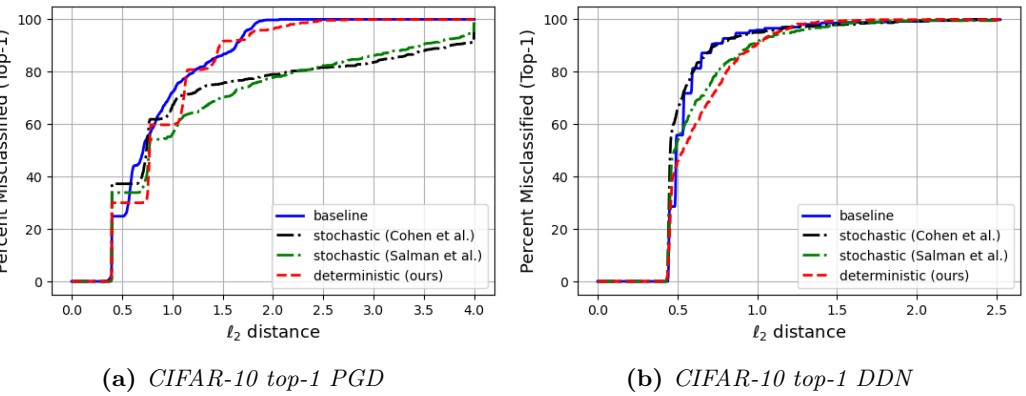

**(a)** *CIFAR-10 top-1 PGD*                    **(b)** *CIFAR-10 top-1 DDN*

**Figure 6:** *CIFAR-10 attack curves: % of images successfully attacked as a function of $\ell_2$ adversarial distance.*

We test our method on the CIFAR-10 dataset (Krizhevsky et al., 2009) with the ResNet-34 model architecture. The initial model $f$ was trained for 200 epochs with the cross-entropy loss function. Our smoothed model $v$ was computed by setting $f^0 = f$ and running Algorithm 1 to minimize the loss (15) with $\sigma = 0.1$ for $n_T = 5$ timesteps at 200 epochs each timestep. The training of our smoothed model took 5 times longer than the baseline model. We compare our results to a ResNet-34 model trained with $\sigma = 0.1$ noisy examples as stochastically averaged model using RANDOMIZEDSMOOTHING (Cohen et al., 2019). We also trained a SMOOTHADV model (Salman et al., 2019) for 4 steps of PGD with the maximum perturbation set to $\varepsilon = 0.5$. To assess certified accuracy, we run the CERTIFY algorithm from Cohen et al. (2019) with $n_0 = 100, n = 10,000, \sigma = 0.1$ for the stochastically trained models. For the HEATSMOOTHING model, we run the same certification algorithm, but without running SAMPLINGUNDERNOISE to compute $\hat{c}_A$. For completeness, we also certify the baseline model $f^0$. Certification plots are presented in Figure 1a. In this plot, we see that our model's $\ell_2$ certified accuracy outperforms the stochastic models. Next, we attack our four models using PGD and DDN. For the stochastic models, we do 100 noise samples to compute the loss. We run both attacks with 20 steps and maximum perturbation $\varepsilon = 4.0$ to force top-1 misclassification. Results are presented in Table 5 and Figures 6a and 6b. In Table 5, we see that HEATSMOOTHING outperforms the stochastic models in terms of robustness. The only exception is robustness to mean PGD perturbations. This is shown in Figures 6a. Our model performs well up to an $\ell_2$ PGD perturbation of just above 1.0.

**Table 5:** *$\ell_2$ adversarial distance metrics on CIFAR-10. A larger distance implies a more robust model.*

| Model | $L$-bound | | PGD | | DDN | |
|---|---|---|---|---|---|---|
| | median | mean | median | mean | median | mean |
| HEATSMOOTHING | 0.094 | 0.085 | 0.7736 | 0.9023 | 0.5358 | 0.6361 |
| SMOOTHADV | 0.090 | 0.078 | 0.7697 | 1.3241 | 0.4812 | 0.6208 |
| RANDOMIZEDSMOOTHING | 0.087 | 0.081 | 0.7425 | 1.2677 | 0.4546 | 0.5558 |
| Undefended baseline | - | - | 0.7088 | 0.8390 | 0.4911 | 0.5713 |

