# OpenReview forum: "Adversarial Boot Camp: label free certified robustness in one epoch"
_ICLR.cc/2021/Conference — Reject_

### Official Review · AnonReviewer3 · 2020-10-25
**Unconvinced results and insufficient discussion on related work**

**Rating:** 4
**Confidence:** 4

**Review:**

This paper proposes to finetune a pretrained network with a gradient norm regularizer to mimic the Gaussian noise augmentation during training. I have two main concerns about this paper:

1. As claimed by the authors, they are motivated to solve the stochasticity of previous certified defenses like random smoothing. However,  there are mainstream certified training methods that are deterministic during inference (with a single query) [1][2][3][4], but are not discussed or empirically compared in this paper.

2. The proposed regularizer function in Eq.(3) is actually equivalent to L-2 adversarial training, which can be simply proved by a dual norm trick [5]. This makes the results in Table 4 seem counter-intuitive, where a from-scratch L-2 adversarially trained model performs worse than a model finetuned by a similar mechanism. I suggest the authors better explain these both formally and empirically.

Minor:
In Sec.4.2, the authors claim that they use the pretrained ImageNet model from Madry GitHub. But as far as I know, there are only pretrained models on MNIST and CIFAR-10 in their repository, could the authors provide an official link to this baseline?



Reference:

[1] Wong et al. Provable defenses against adversarial examples via the convex outer adversarial polytope. ICML 2018

[2] Wong et al. Scaling provable adversarial defenses. NeurIPS 2018

[3] Dvijotham et al. Training verified learners with learned verifiers. arXiv 2018

[4] Dvijotham et al. A dual approach to scalable verification of deep networks. UAI 2018

[5] Simon-Gabriel et al. First-order adversarial vulnerability of neural networks and input dimension. ICML 2019

---

### Official Review · AnonReviewer1 · 2020-10-27
**Official Blind Review #1**

**Rating:** 3
**Confidence:** 4

**Review:**

The paper claims that a (computationally intractable) randomized smoothing of any classifier can be distilled into the (deterministic) classifier itself via fine-tuning it with gradient penalty. This is motivated by a theoretical result that Gaussian smoothing of a classifier is equivalent to solving a certain heat equation, which can be approximated by a regularized loss training. Experimental results use the resulting deterministic classifier to compute the certified radius compared to (stochastic) smoothed classifiers, arguing its efficiency and higher certified radius of the proposed method.

The relation between randomized smoothing and PDE seems to be an interesting direction to explore. My biggest concern is that, however, whether the proposed deterministic smooth classifier is indeed *certifiably* robust in practice. Apart from its theoretical motivation, I could not find any statistical guarantee that the deterministic smooth classifier is provably close to its stochastic counterpart for every input x, which is essential to claim the certifiable robustness of the proposed model. Otherwise, how one could prove that the deterministic smooth classifier is indeed robust to adversarial examples? - although randomized smoothing may require much more computation time, it at least provides such a statistical yet practical guarantee, namely as the CERTIFY algorithm [1].

Another possible way to go is to show that the *empirical* robustness of the deterministic smooth classifier is non-trivial. In this context, actually, the message of the paper can be "very" surprising (and very unlikely, at the same time): one can robustify any classifier by a single pass of fine-tuning with gradient penalty, even without adversarial training. The paper indeed provide a related result, i.e., empirical test accuracy on adversarial attacks, but the current evaluation is too weak to support the claim: considering a bunch of defense papers that are broken after published [2, 3], it is too hard for me to believe the results as is even it could bear from a specific configuration of PGD and DDN. I would recommend the paper to follow the standard guidelines suggested by [4]. The paper could explore more possible configurations of PGD attacks, or other types of attacks (e.g., gradient-free attacks [5], or black-box attacks), just to name a few.

[1] Cohen et al., Certified Adversarial Robustness via Randomized Smoothing, ICML 2019.

[2] Athalye et al., Obfuscated Gradients Give a False Sense of Security: Circumventing Defenses to Adversarial Examples, ICML 2018.

[3] Tramer et al., On Adaptive Attacks to Adversarial Example Defenses, 2020.

[4] Carlini et al., On Evaluating Adversarial Robustness, 2019.

[5] Uesato et al., Adversarial Risk and the Dangers of Evaluating Against Weak Attacks, ICML 2018.

---

### Official Review · AnonReviewer2 · 2020-10-27
**This paper propose a deterministic method for certified robustness under adversarial attack. Different from the prior random smoothing approach, this paper uses deterministic inference and achieves times of speed up. The idea is novel and interesting.**

**Rating:** 7
**Confidence:** 3

**Review:**

Randomized smoothing is the major way to certify the robustness of large scale networks, however, it requires sampling from Gaussian distribution many times, which is not fast enough for real-time inference. This paper uses a regularized loss to get deterministic Gaussian averaged results. This paper points out an interesting direction for certifying robustness, the method is simple and effective.

Strength:

1. The paper is clearly written.

2. It is very interesting to see a method that can certify the robustness without the computational intensive randomized smoothing.

3. The method is simple and effective, does not require much computation resources, which improves the inference speed by around 10 times (Table 2).

4. The speed for randomized smoothing certification is a major concern for the community. This paper address this problem. If this paper is really effective, it can have a broad impact on the research community for robustness certification.

Weakness and Questions:

1. What if the attacker directly attacks the objective function for certification (e.g. equation 2) ? Given that the certification is deterministic, is it possible to fool the certification method?

2. In Figure 2, it seems the deterministic under performs some baselines.

---

### Official Review · AnonReviewer4 · 2020-10-28
**Problems with the proposed approach**

**Rating:** 3
**Confidence:** 4

**Review:**

This paper proposes to use a deterministic classifier to replace the sampling
process in randomized smoothing based certifiably robust models.  The goal of
training a deterministic robust classifier to avoid the high cost of randomized
smoothing is a right direction to look at.

The reason that we can get certified robustness with Gaussian smoothing is that
the smoothed classifier becomes Lipschitz (see Salman et al.). Unfortunately,
it is a stochastic classifier and typically it is impossible to access the
smoothed classifier directly, and that's why in randomized smoothing (e.g.,
Cohen's PREDICT procedure), sampling is necessary.

The approach in this work is to train a smoothed classifier that essentially
returns the same prediction as the mean of the originally stochastic
classifier. The authors pointed out the connection between Gaussian smoothed
classifier and gradient regularization, so they use gradient regularization to
obtain the desired deterministic classifier.

Unfortunately, it seems to me that the proposed approach is not sound. With
gradient regularization, we try to make the learned classifier to be smooth,
however there is no guarantee that such a learned classifier will indeed be
smooth and produce the same outcome as the original Gaussian smoothed
classifier, so we cannot use the outcome of this classifier to replace Cohen's
SampleUnderNoise procedure.  More precisely, optimizing the proposed loss
function (3) does not guarantee the learned classifier f^smooth to be Lipschitz,
where the original Gaussian smoothed classifier is guaranteed to be Lipschitz.
Although the loss attempts to do so with gradient regularization, there is no
guarantee here. So the entire procedure is not certified anymore.

On the positive side, the proposed method may work as a good empirical defense,
since the smoothed classifier can be learned quickly and can be more robust
than the original classifier. This may be advantageous for certain applications
where adversarial training is too slow or we don't want to retrain the original
classifier.

Because of the fundamental problem mentioned above, I cannot recommend
acceptance of this paper. I am willing to discuss with the authors further in
case I misunderstand some parts of the paper.

---
### After rebuttal:

After reading the rebuttal, I feel my main concern is still not addressed by the response. The authors agree that they may provide a different kind of guarantee for the certificates as in (Cohen et al., Salman et al.). An empirical comparison between the output of the proposed model and the mean from sampling is not sufficient. We hope the authors can improve on this point and provide formal robustness guarantees like those in (Cohen et al., Salman et al.).

---

### Author Response · Authors · 2020-11-24
**Rebuttal to Reviewer Comments and Submission of Rebuttal Revision**

Thank you reviewers for taking the time to give helpful feedback on our paper. In this work, we aimed to use ideas from PDEs in order to develop a training scheme that results in a Gaussian average of some initial model. In doing so, we found it possible to take any baseline computer vision model and make it certifiably robust to input perturbations. While perhaps not as robust as other certification methods, the main goal of this work was to use PDE theory and calculus of variations to solve the heat equation. Since the solution of the heat equation is in theory equivalent to Gaussian smoothing, we let our main application be computer vision to draw links between our work and that of Cohen et. al. With this in mind, future work must be done to show that we can solve other PDEs using our iterative training method. As pointed out, there were some shortcomings of this work that we wish to address.

It is indeed a good point that our model may not follow the same certification guarantees as Cohen's stochastically averaged models. In an attempt to show that our model is indeed averaged, we added a section to our revised submission. In section 4.3, we first compare how well our deterministic model's prediction matches predictions of this same model obtained using the stochastic ``voting'' method from Cohen et. al. PREDICT algorithm. We then see how often our model's deterministic single-forward-pass predictions matches the predictions of a model from Cohen et. al. Given that both percentages of agreement in predictions are high, we do indeed believe that our model is worthy of being considered a Gaussian average of some initial model, thus allowing us to implement the algorithms presented in Cohen et al. and Salman et al.

The point was made that our approach is equivalent to L2 regularization and it therefore not novel. We note that we aren't doing L2 Loss regularization, we are regularizing the vector output of the model. This approach is more aligned to the PDE theory. Furthermore, we implement numerical approximation methods to compute the L2 norm of the Jacobian matrix. To our knowledge, this is not previously done in papers that use regularization as an adversarial defense. Furthermore, a reviewer pointed out that the results in Table 4 are counter-intuitive. The purpose of Table 4 is to show that an adversarially trained model (from Madry et al.) doesn't have a certified radius in terms of the stochastic procedure presented in Cohen et al. By running our ImageNet training procedure, we showed that we can perform Gaussian smoothing on an adversarially trained model, thus giving it a certified radii.

One reviewer made the very good point that we could have implemented other adversarial attacks and compare our deterministic model's performance against known defense methods such as adversarial training. In fact, we tried this and came across two limitations in doing so. One, we lacked the time and computational resources to implement several state-of-the-art attacks on ImageNet. Secondly, we found that our model under-performed compared to adversarial training. This came as no surprise. In our work we never tried to provide the best adversarial defense. Rather, we demonstrate that using PDE theory, we can obtain models equivalent to the stochastic models in Cohen et al. It is work nothing that in practice, Cohen et al.'s models do not beat adversarial training.

A question was brought up as to whether or not there are PGD adversarially-trained ImageNet models available to download online. There are indeed pretrained ImageNet ResNet-50 models in Aleksander Madry's ``robustness'' GitHub repository. Download links are available towards the end of the following README.md file https://github.com/MadryLab/robustness/blob/master/README.rst.


For references to any papers mentioned in this comment, please see the "References" section of our paper.

---

### Decision · Program_Chairs · 2021-01-07
**Final Decision**

**Decision:**

Reject

**Comment:**

Although the connection between randomized smoothing and PDE revealed in this paper is an interesting direction to explore, the method proposed unfortunately is not certified. The method could work as a good empirical defense since the smoothed classifier could be learned more efficiently.